# Complement Activation Products in Patients with Chronic Schizophrenia

**DOI:** 10.3390/jcm12041577

**Published:** 2023-02-16

**Authors:** Krzysztof Rudkowski, Katarzyna Waszczuk, Ernest Tyburski, Katarzyna Rek-Owodziń, Piotr Plichta, Piotr Podwalski, Maksymilian Bielecki, Monika Mak, Anna Michalczyk, Maciej Tarnowski, Katarzyna Sielatycka, Marta Budkowska, Karolina Łuczkowska, Barbara Dołęgowska, Mariusz Z. Ratajczak, Jerzy Samochowiec, Jolanta Kucharska-Mazur, Leszek Sagan

**Affiliations:** 1Department of Psychiatry, Pomeranian Medical University, Broniewskiego 26, 71-460 Szczecin, Poland; 2Department of Health Psychology, Pomeranian Medical University, Broniewskiego 26, 71-460 Szczecin, Poland; 3Department of Physiology in Health Sciences, Pomeranian Medical University, Żołnierska 54, 70-210 Szczecin, Poland; 4Institute of Biology, Faculty of Exact and Natural Sciences, University of Szczecin, Felczaka 3c, 71-415 Szczecin, Poland; 5Department of Medical Analytics, Pomeranian Medical University, Powstańców Wielkopolskich 72, 70-111 Szczecin, Poland; 6Department of General Pathology, Pomeranian Medical University, Powstańców Wielkopolskich 72, 70-111 Szczecin, Poland; 7Department of Laboratory Medicine, Pomeranian Medical University, Powstańców Wielkoposlkich 72, 70-110 Szczecin, Poland; 8Stem Cell Institute, James Graham Brown Cancer Center, University of Louisville, Louisville, KY 40292, USA; 9Department of Neurosurgery, Pomeranian Medical University, 71-252 Szczecin, Poland

**Keywords:** complement cascade, schizophrenia, C3a, C5a, C5b-9, PANSS

## Abstract

Evidence suggests a role of the immune system in the pathogenesis of a number of mental conditions, including schizophrenia (SCH). In terms of physiology, aside from its crucial protective function, the complement cascade (CC) is a critical element of the regeneration processes, including neurogenesis. Few studies have attempted to define the function of the CC components in SCH. To shed more light on this topic, we compared the levels of complement activation products (CAP) (C3a, C5a and C5b-9) in the peripheral blood of 62 patients with chronic SCH and disease duration of ≥ 10 years with 25 healthy controls matched for age, sex, BMI and smoking status. Concentrations of all the investigated CAP were elevated in SCH patients. However, after controlling for potential confounding factors, significant correlations were observed between SCH and C3a (M = 724.98 ng/mL) and C5a (M = 6.06 ng/mL) levels. In addition, multivariate logistic regression showed that C3a and C5b-9 were significant predictors of SCH. There were no significant correlations between any CAP and SCH symptom severity or general psychopathology in SCH patients. However, two significant links emerged between C3a and C5b-9 and global functioning. Increased levels of both complement activation products in the patient group as compared to healthy controls raise questions concerning the role of the CC in the etiology of SCH and further demonstrate dysregulation of the immune system in SCH patients.

## 1. Introduction

Affecting about 1% of the general population [1], schizophrenia (SCH) is a complex, chronic mental disorder, whose manifestation includes such symptoms as delusions, hallucinations, social withdrawal and blunted affect, among others [2]. Even though SCH is probably the most frequently and widely investigated neuropsychiatric disorder, its biological background is yet to be established.

The neurodevelopmental theory of SCH, formulated by Weinberger, Murray and Lewis [3,4], provided a significant foundation for understanding a syndrome that typically appears with severe pathology in adolescence or early adulthood as a result of events that occurred early in development. However, this theory does not account for all the biological symptoms observed in its course. Therefore, further hypotheses to explain its underlying etiopathogenesis are underway.

As early as in the 1980s, the immune hypothesis was formulated, which assumes that the immune response and subsequent inflammation within the nervous system may lead to progressive brain changes in patients suffering from SCH [5,6]. The possibility that the immune system plays a role in chronic illnesses has been gaining increasing attention in recent years. Epidemiological studies suggest such immunological involvement, providing evidence for associations between schizophrenia risk and autoimmune diseases, prior infection-related hospitalizations, as well as prenatal and childhood illnesses [7,8]. Likewise, research shows higher levels of inflammatory markers TNF-α, IL-6, IL-1, sIL-2R, and C-reactive protein in peripheral blood of individuals with first-episode psychosis and SCH [9,10,11]. In addition, cerebrospinal fluid (CSF) analyses have provided ample evidence that inflammatory processes in the central nervous system (CNS) are involved in the pathogenesis of SCH. According to a 2018 meta-analysis, patients with SCH had significantly higher levels of IL-6 and IL-8 in their CSF compared to healthy controls [12]. When considered collectively, the evidence supports the theory that schizophrenia, at least in some individuals, may be a neuroimmune-mediated condition caused by changes in pro- and anti-inflammatory processes in the CNS [13].

The complement system is composed of plasma proteins that are involved in the elimination of cellular waste products and defense against invading infections [14]. Each of the three key pathways of the complement cascade (CC) (i.e., classical, lectin and alternative) is activated by different mechanisms. However, all the pathways lead to a formation of C3 convertase, generating its active fragments C3a and C3b, a formation of C5 convertase and release of C5a and C5b fragments and, finally, a formation of a membrane attack complex MAC (C5b-C9), which causes cell lysis [15]. By attracting phagocytes and inducing vasodilation, as anaphylatoxins, C3a and C5a mediate inflammation.

In addition to its involvement in innate and adaptive immunity, CC is crucial for synaptic pruning in the brain, both during development and pathological conditions, including neuroinflammatory diseases [16]. Fragments of complement cleavage C3a and C5a are anaphylatoxins that stimulate chemotaxis and leucocyte activation to regulate the local pro-inflammatory response causing the release of IL-1, IL-6 and vasoactive amines, which enhance vascular permeability and cause vasodilatation [17]. Additionally, C3a plays a crucial role in the activation of leukocytes and endothelial cells within the CNS [18]. Moreover, it has been demonstrated that complement cleavage fragments may affect the blood–brain barrier (BBB), which in healthy persons inhibits plasma proteins from entering the CNS [18,19]. Leakage of the BBB causes microglial activation and cytokine release that affects neurogenesis, impacts neurotransmitter and white matter function and may also worsen SCH symptoms including cognitive decline [20,21]. Therefore, increased complement activation products (CAP) levels may support the hypothesis that inflammation plays a role in the etiology of SCH.

On the other hand, it has been reported that under hypoxic conditions, C3a stimulates the differentiation of neural progenitor cells and, thus, has an impact on neurogenesis [22]. For instance, the lectin pathway is essential for normal neuronal migration during neurodevelopment [23]. Evidence suggests that C3 deficient mice have defective neuronal migration, which impacts postnatal cortical organization [24]. Additionally, C3a alters the response of astrocytes to ischemia by enhancing their capacity to withstand ischemia-related stressful conditions. What is more, it functions as a chemo-attractant for neural cells during antenatal development [15,22,25]. In the presence of microglia inhibitor minocycline [26], complement receptor 3 (CR3; CD11b/CD18) binds with C3 degradation fragment iCb3 [27]. Interestingly, disruption in microglia-specific CR3/C3 signaling has been linked with long-term synaptic connection impairments [27]. These findings suggest that microglia have the potential to interact with synapses to support activity-dependent plasticity, inducing retinogeniculate input engulfment, which is a likely underlying mechanism of synaptic pruning in developing neural circuits.

Nevertheless, the involvement of the complement system in the pathogenesis of SCH remains unclear. Still, it seems that it is rather genetic studies and not the levels or activity of the complement products that provide the majority of the available evidence on such links. Genome-wide association studies (GWAS) demonstrated a genetic association with SCH for multiple loci [28]. As recently reported by Woo et al. [29], genetic investigations have found multiple complement-related risk loci for SCH. The strongest genetic link for SCH is with genetic markers located in the major histocompatibility complex (MHC) locus [30] which comprises four genes associated to the complement system (complement factors B, C2, C4A and C4B) [29]. Following this discovery, researchers examined post-mortem brain samples and discovered elevated C4A mRNA in the brains of SCH individuals. C4 was also discovered in synapses in the prefrontal cortex (PFC) and the hippocampus of human brain samples, suggesting that it may be involved in synaptic pruning, which may contribute to the lower synapse number observed in SCH patients [30]. Combining their results on the link between SCH and C4A with previous evidence on C1q-mediated postnatal synaptic pruning, [31] offered an important perspective on the involvement of the complement system in SCH.

Non-genetic evidence is more scarce and inconclusive. C4A and C4B were both found to be increased in SCH individuals in subsequent research [29]. In a different study, individuals with first episode psychosis (FEP) and chronic SCH had different levels of complement components. Elevated C4 and lower C3 levels were seen in patients with a longer history of therapy, while patients with FEP had higher concentrations of both components [32]. In addition, it has been demonstrated that the complement activation product C3a may serve as a risk indicator and a possible FEP marker [33]. Föcking et al. [34] recently conducted a longitudinal study in a cohort of teenagers and found evidence linking psychotic episodes at age 18 with elevated levels of various complement proteins detected years earlier at age 12.

The main pathological findings in the brains of SCH patients are (i) fewer dendritic spines on prefrontal cortex neurons and (ii) a severe loss of grey matter [35,36,37]. The identification of C4 as a schizophrenia-associated genetic variant prompted evaluation of the complement protein mRNA expression and its correlation with cortical thickness in SCH. Interestingly, a reduced superior frontal cortex thickness was shown to be correlated with increased C5 expression [38]. Similar to C3a, C5a causes chemotaxis that promotes recruitment of microglia as a result of decreased synaptic activity. The increased concentration of C5 may be linked to reduced cortical thickness [39].

Although not widely investigated, there is a likely immunomodulatory effect that antipsychotic medication may have on the activation of the complement system [40,41]. Nevertheless, research stratified by medication status yields conflicting results. For example, Boyajyan et al. reported higher C3 hemolytic activity in patients taking antipsychotic drugs compared to individuals who had never been on antipsychotics [42]. What is more, Zhang et al. observed increased serum mRNA expression of C3 in SCH patients with clozapine-induced metabolic syndrome compared to clozapine-taking patients without the metabolic syndrome [43]. Based on these findings, it is probable that prior reports of increased C3 and C4 concentrations in patients versus controls were induced partly by medication-induced weight gain in the patient group. These findings, however, should be regarded with caution, since two additional studies found no significant change in the serum complement concentrations or complement activity levels between medicated and non-medicated individuals [44,45].

However, the available literature does not provide comprehensive information on possible changes in complement concentrations in chronic schizophrenia. No research to date addressed alterations in active cleavage forms of CC in patients with chronic SCH. Therefore, the aim of this study was to shed more light on this matter and (i) evaluate alterations in the levels of complement activation products (CAP)—C3a, C5a and C5b-9—in patients with chronic SCH and healthy controls as well as (ii) determine whether these alterations are related to the psychopathological manifestation of SCH. Based on the existing literature, we hypothesized that (i) the level of CAP in the peripheral blood will differ between SCH and HC group and (ii) SCH symptom severity will correlate with those altered levels.

## 2. Materials and Methods

### 2.1. Participant and Procedure

For this study, 62 unrelated patients suffering from SCH were recruited from inpatient, day treatment and outpatient facilities at the Department of Psychiatry of the Pomeranian Medical University in Szczecin, Poland. Prior to enrollment, all participants provided their informed consent to the study (approved by the Ethics Committee of the Pomeranian Medical University in Szczecin KB-0012/159/17 from 18 December 2017). The inclusion criteria were as follows: diagnosis of schizophrenia according to ICD-10 [46], duration of illness of ≥10 years, age of 30–50 years, absence of substance use disorder (SUD) and serious somatic comorbidity, especially of inflammatory origin. The control group consisted of 25 healthy volunteers, without substance use or other psychiatric disorders nor serious somatic comorbidity, especially inflammatory diseases.

A standard physical, neurological and mental state examination, including laboratory blood tests, was performed by a trained psychiatrist. A standardized original self-report questionnaire was used to collect data on demographics and family history. The Mini International Neuropsychiatric Interview (MINI) was used to rule out the presence of any psychiatric illnesses other than SCH [47]. The Positive and Negative Syndrome Scale (PANNS), a standardized tool for the multidimensional assessment of patients with SCH, was used to evaluate the severity of psychopathological symptoms. We adopted Shafer and Dazzi’s five-factor solution to analyze the PANSS results (i.e., positive, negative, cognitive/disorganized, depression/anxiety, and hostility symptoms) [48]. None of the patients were in acute psychosis at the time of the study. To evaluate the general patient functioning, we used the Global Assessment of Functioning (GAF) [49]

We used a chlorpromazine equivalent, which is a dosage of antipsychotic medications with a similar strength to 100 mg of chlorpromazine, to compare the daily medication doses taken by the patients [50]. The attending psychiatrist decided on the medication and the dosage. The recommended course of treatment was not altered in any way by the research team. Healthy controls underwent similar examinations to exclude mental disorders and somatic diseases.

### 2.2. Measurement of Complement Cascade Components

After an overnight fast, venous blood samples were taken between 8 and 9 in the morning. For our purposes, we used EDTA plasma that was centrifuged (3000 rpm, 10 min, 4 °C), separated into Eppendorf tubes and frozen at −80 °C until assayed (i.e., for 3 months). After a single thawing, all measurements were performed. We used the Human C5b-9 ELISA Set, Catalog No 558315, to measure the C5b-9 (MAC) concentration (BD OptEIA), while the concentrations of C3a and C5a were determined using the Human C3a ELISA Kit, Catalog No 550499 and the Human C5a ELISA Kit, Catalog No 557965 (BD OptEIA).

### 2.3. Statistical Analysis

Statistical analysis was performed using IBM SPSS 27 (IBM Corp., Redmont, VA, USA). Continuous variables were presented as means (M) and standard deviations (SD). The normality of distributions, skewness and kurtosis were tested with the Shapiro–Wilk test. We assumed that skewness values of −2 to +2 and kurtosis values of −7 to +7 indicated normal distribution of variables [51]. Age in both groups, as well as global functioning measured with GAF and chlorpromazine equivalent in the clinical group were normally distributed. Years of education, BMI, biological parameters, symptom severity measured with the PANSS, illness duration and exacerbation were not normally distributed. Differences between the two groups in terms of age were examined with Student’s *t* test (if the relevant assumptions were met), while in terms of years of education, BMI and complement activation products (C3a, C5a and C5b-9) with the Mann–Whitney *U* test (if the relevant assumptions were not met). In addition, in the case of significant differences in the biological parameters between the two groups, we conducted an analysis of covariance (ANCOVA) to control for the effects of age, years of education and BMI. Cohen’s *d* (for Student’s *t* test) and ɳ^2^ (for ANCOVA) [40] and Wendt’s r_u_ (for Mann–Whitney *U* test) [52] were used to determine the magnitudes of effect sizes for inter-group differences. Finally, in order to assess the relationships between demographic or clinical variables, BMI or psychopathology and biological parameters, Spearman’s *rho* correlation coefficient was estimated. The Holm–Bonferroni *p*-value correction was used for all statistical analyses with multiple comparisons and correlations. Additionally, multivariate logistic regression (backward elimination, the Wald chi-square test) was further performed to identify independent biological risk factors of schizophrenia (with the Hosmer–Lemeshow test for evaluating the goodness of fit of logistic regression models). The alpha criterion level was set at 0.05 for all statistical analyses.

## 3. Results

### 3.1. Participant Characteristics

Demographic, clinical characteristics and BMI are presented in Table 1. There were no significant group differences in terms of sex. However, as compared to healthy controls, SCH patients were older (*p*-corrected = 0.008), reported fewer years of education (*p*-corrected = 0.046) and had higher BMI (*p*-corrected = 0.014).

### 3.2. Differences in Complement Activation Products

As presented in Table 2 and Figure 1, SCH patients had higher C3a (*p*-corrected < 0.00001), C5a (*p*-corrected = 0.02) and C5b-9 levels (*p*-corrected = 0.049) than healthy controls. Of note, differences in C3a and C5a levels remained significant after adjusting for age, years of education and BMI (*F*_(1, 78)_ = 9.78; *p* = 0.002; ɳ^2^ = 0.11 and *F*_(1, 78)_ = 4.18; *p* = 0.044; ɳ^2^ = 0.05). In addition, multivariate logistic regression showed that C3a (*p* < 0.001) and C5b-9 (*p* = 0.033) were significant predictors of schizophrenia, explaining about 47% of the variance (the model showed good fit to the analyzed data, *H* = 34.46; *p* < 0.001).

### 3.3. Complement Activation Products, Psychopathology and Global Functioning

After the Holm–Bonferroni *p*-value correction, there were no significant Spearman’s *rho* correlations between any CAP and SCH symptom severity (i.e., positive, negative, cognitive/disorganization, depression/anxiety or hostility symptoms) or general psychopathology in SCH patients. However, two significant positive Spearman’s *rho* correlations emerged between (i) C3a and global functioning (*rho* = 0.46; *p*-corrected < 0.001) and (ii) C5b-9 and global functioning (*rho* = 0.41; *p*-corrected <0.001). All correlations are presented in Table 3.

## 4. Discussion

This study compared the levels of complement activation products (CAP) (C3a, C5a and C5b-9) in patients with chronic schizophrenia (SCH) and healthy controls (HC). We demonstrated significantly higher concentrations of C3a and C5a in patients with SCH as compared to HC. To our knowledge, this is the first study to assess CAP among patients with chronic SCH. Given the relative paucity of research addressing alterations in CAP in patients with chronic SCH, it is difficult to compare our findings to available data in the field. Indeed, most of the current research does not assess the same active cleavage forms of CC that are central to this study [53]. What is more, available reports tend to demonstrate inconsistent results. Despite documented evidence that numerous CC dysregulations do occur in SCH [54], a recent meta-analysis found no significant differences in the levels of C3 or C4 between SCH patients and HC [53]. Likewise, while some studies revealed significantly higher levels of C3 concentrations in SCH patients as compared to HC [55,56,57], others found significantly lower levels of C3 in SCH [58,59] or no differences between the same patient population and controls [60,61,62]. Laskaris et al. observed higher blood C3 and C4 levels in ultra-high risk (UHR) patients and increased C4 level among patients with SCH as compared to HC [63].

Of particular note, not only did our analyzes reveal the elevated concentrations of C3a and C5a in the clinical group, but, more importantly, they indicated that C3a and C5b-9 may in fact predict SCH. This, therefore, further suggests that the latter two complement activation products may be construed as potential biological markers of SCH and that, if so, they could be useful (at least as a diagnostic adjunct) in the identification of patients with (a chronic course of) SCH. Nonetheless, this hypothesis should be approached with great caution, as (i) this research could be considered a pilot study striving to search for biological underpinnings of schizophrenia and, therefore, our results should be interpreted as only preliminary and (ii) there exist factors this study did not control for that could confound the demonstrated relationship, thus warranting further exploration of this matter.

Although relatively lacking in the population of SCH patients with a long history of treatment, there exist reports on the concentrations of CAP (C3a, C5a and C5b-9) among other psychotic patient groups with different duration of illness. Indeed, there is evidence of significantly reduced C3a in FEP patients as compared to HC, with no significant differences reported for either C5a or C5b-9 [33]. In contrast, elsewhere [64], increased levels of C3a were found in UHR and FEP groups, both relative to HC. Due to the existing inconsistencies, it seems worthwhile to further investigate CAP alterations as not only relative to healthy controls but also across various SCH presentations along the entire continuum of schizophrenia spectrum disorders (UHR vs. FEP vs. chronic SCH).

Alterations of CC concentrations in peripheral blood constitute an integral part of the immune hypothesis of SCH. In particular, the sterile inflammation theory formulated by Ratajczak et al. [65], linking chronic inflammation to psychiatric disorders offers an interesting explanation of our results. According to its postulates, CC activation does not necessarily require the presence of any external pathogen but may be triggered instead by an increase of danger-associated molecular pattern (DAMP) mediators in the brain, including extracellular ATP and high-mobility group box 1 (HMGB1) protein. The release of DAMPs from the brain tissue activates microglia and astrocytes or, from circulating innate immune cells, is involved in the activation of complement cascade in a MBL-dependent manner. While there is evidence to support this notion with respect to various psychiatric disorders [66], the concept of sterile inflammation perpetuating the activation of CAP and its exact mechanism underlying psychosis seems a viable proposal for future research.

Listed among the important candidates that are likely to affect the levels of CC proteins are antipsychotic drugs. In our study, the dose of the administered pharmacological treatment (expressed via chlorpromazine equivalent) did not affect the results. However, what needs to be stressed is that we only focused on the dosage and not the medical status or specific drugs used. Existing research concerning the effect of antipsychotic medication on CC levels yields conflicting results, reporting either increased complement activity in medicated patients compared to controls or antipsychotic-naive individuals, or report no overall relevance of drugs [67]. Such variability of available research results implies a likely effect of medication-related or medication-induced factors that may, at least potentially, underpin complement activation mechanisms in SCH patients and should, therefore, be included in future research paradigms.

Similar inconsistencies are also found across studies that compare plasma C3 and C4 levels and SCH symptoms. While no significant correlations between C3, C4 and PANSS scores were observed in some studies [55,59], another one reported a link between serum C3 level and the social withdrawal/passive apathy subscale of the PANSS [68]. In our study, we did not find any significant correlations between CAP and psychopathological presentation, although a significant positive correlation emerged between C3a and C5b-9 components and the Global Assessment of Functioning (GAF). This seems a particularly interesting finding, especially in terms of the elevated CAP as the potential biological markers of SCH, raising an important question: what aspects of SCH are predicted by the presence of pro-inflammatory markers if it is not the severity of psychopathological presentation? If not anything else, this seems to be a good starting point for future scientific enquiries.

In the available literature, there are no data on the relationship between the GAF scale and CC. The significant positive correlation between C3a and C5b-9 components and the GAF scale we observed in this study can, on the one hand, be interpreted as their effect on the subjective report of social, occupational and psychological functioning and, on the other, a reflection of an aspect of psychopathological manifestation, yet a different one from that examined by the PANSS. The latter interpretation would be much in line with what has been proposed in other studies with the use of the GAF scale [69]. Overall, especially if considered to play a role in the natural progression of SCH, the observed relationship could suggest that CAP levels influence the subjective perception of quality of life and self-reported well-being in SCH patients with a long history of illness.

The causes underlying the above general data inconsistencies are uncertain, although they are attributable to a number of reasons including differences in terms of applied molecular techniques, medication status, methodological factors or clinical heterogeneity of patients, thus making further investigation of the postulated effect of complement activation on the course of schizophrenia a timely and valid endeavor.

Although the undoubted novelty of this study consists in the insights it offers into the presentation of chronic schizophrenia as there are no data to date concerning complement activation products in the course of chronic SCH, there are two significant limitations that should be considered when interpreting our data. Our research was carried out in a single center on an entirely Polish and relatively small cohort. Therefore, to enhance generalizability of results, a large-scale multicenter replication thereof on a larger and more heterogeneous sample is required to further support our findings. What is more, as the abnormalities in CC levels in SCH can reflect a dynamic process that depends on the phase of the disease, it seems relevant to investigate the effect of its duration, which in several studies is estimated with limited precision based on self-reported data. Given the reported differences in complement activation product levels in patients at different stages of illness (UHR, FEP, chronic SCH), a prospective study of likely changes in those compound levels over the course of SCH seems warranted.

## 5. Conclusions

In conclusion, our research showed increased levels of C3a and C5a in SCH patients as compared to healthy controls, and C3a and C5b-9 were significant predictors of schizophrenia. Contrary to expectations, we did not find significant correlations between the three investigated complement activation products and psychopathological presentation, although there were links between C3a and C5b-9 components and the general self-reported level of patient functioning. Our findings, therefore, add to the existing body of evidence suggesting that CC may play a role in the course of SCH, thus, further supporting the notion that the immune system and regeneration processes are disturbed in SCH. Apart from etiological insights, complement system dysregulations may offer clinical utility as potential biomarkers of schizophrenia. Our results, however, need to be replicated on larger patient populations.

## Figures and Tables

**Figure 1 jcm-12-01577-f001:**
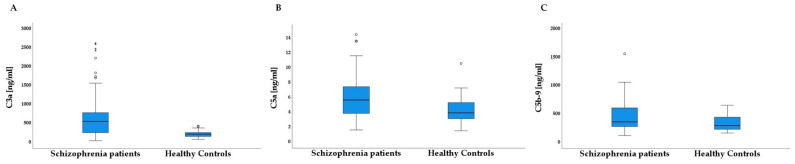
Comparisons of the mean number of complement components between schizophrenia patients and healthy controls ((**A**) = for C3a; (**B**) = for C5a, and (**C**) = for C5b-9). In all box plots, the bottom end of the box designates the first quartile, a line within the box indicates the median and the top end of the box shows the third quartile. Whiskers indicate values 1.5 times the interquartile range below the first quartile and above the third quartile. Crosses represent average values. Circles designate individual observations.

**Table 1 jcm-12-01577-t001:** Demographic and clinical characteristics and BMI of schizophrenia patients and healthy controls.

Variable	Schizophrenia Patients (*n* = 62)	Healthy Controls (*n* = 25)	*t*/*Z*/*χ*^2^	*p*/*p*-Corrected	*d*/*r_U_*
Age: *M* (*SD*)	39.54 (6.78)	34.08 (8.08)	3.11 ^a^	0.003/0.008	0.67 ^d^
Years of education: *M* (*SD*)	13.28 (2.79)	14.60 (2.47)	−1.99 ^b^	0.046/0.046	0.28 ^e^
Sex: female/male	24/38	13/12	1.29 ^c^	0.257/-	-
BMI: *M* (*SD*)	28.01 (4.76)	24.97 (3.59)	−2.71 ^b^	0.007/0.014	0.46 ^e^
Antipsychotic medications:					
Atypical: *n* (%)	43 (69.40)	-	-	-	-
Atypical and typical: *n* (%)	16 (25.80)	-	-	-	-
Typical: *n* (%)	2 (3.20)	-	-	-	-
No medications: *n* (%)	1 (1.60)	-	-	-	-
Chlorpromazine equivalent (mg): *M* (*SD*)	686.19 (298.76)	Min-Max = 0.00–1500.00			
Duration of illness: *M* (*SD*)	15.32 (5.50)	Min-Max = 10–28			
Exacerbation: *M* (*SD*)	6.26 (3.70)	Min-Max = 1–24			
Global functioning in GAF: *M* (*SD*)	55.64 (14.70)	Min-Max = 25–88			
Positive Symptoms in PANSS: *M* (*SD*)	7.89 (3.89)	Min-Max = 5–22			
Negative Symptoms in PANSS: *M* (*SD*)	17.45 (6.43)	Min-Max = 7–34			
Cognitive/Disorganization in PANSS: *M* (*SD*)	12.23 (3.95)	Min-Max = 8–27			
Depression/anxiety in PANSS: *M* (*SD*)	8.66 (3.41)	Min-Max = 5–18			
Hostility in PANSS: *M* (*SD*)	4.74 (2.10)	Min-Max = 4–19			
Psychopathology in PANSS: *M* (*SD*)	53.10 (13.10)	Min-Max = 31–109			

GAF = Global Assessment of Functioning. PANSS = Positive and Negative Syndrome Scale. ^a^ Student’s *t* test. ^b^ Mann–Whitney *U* test. ^c^ Chi-squared test. ^d^ Cohen’s *d* effect size: small (0.20–0.49), medium (0.50–0.79), large (>0.80). ^e^ Wendt’s *r* rank-biserial correlation effect size: small (0.10–0.29), medium (0.30–0.49), large (>0.50).

**Table 2 jcm-12-01577-t002:** Comparisons of the mean levels of complement components between schizophrenia patients and healthy controls.

	Schizophrenia Patients (*n* = 62)	Healthy Controls (*n* = 25)	*Z*	*p*/*p*-Corrected	*r_U_*
C3a [ng/mL]: *M* (*SD*)	724.98 (663.80)	226.22 (100.70)	−4.44 ^a^	0.000/0.000	0.61 ^b^
C5a [ng/mL]: *M* (*SD*)	6.06 (2.91)	4.39 (2.00)	−2.58 ^a^	0.010/0.020	0.35 ^b^
C5b-9 [ng/mL]: *M* (*SD*)	454.71 (264.19)	343.74 (153.67)	−1.97 ^a^	0.049/0.049	0.27 ^b^

^a^ Mann–Whitney *U* test. ^b^ Wendt’s *r* rank-biserial correlation effect size: small (0.10–0.29), medium (0.30–0.49), large (>0.50).

**Table 3 jcm-12-01577-t003:** Correlations between complement components, psychopathological dimensions and global functioning in schizophrenia patients.

Variable	Positive Symptoms in PANSS		Negative Symptoms in PANSS		Cognitive/ Disorganization in PANSS		Depression/ Anxiety in PANSS		Hostility in PANSS		Psychopathology in PANSS		Global Functioning in GAF	
	*rho*	*p*-Corrected	*rho*	*p*-Corrected	*rho*	*p*-Corrected	*rho*	*p*-Corrected	*rho*	*p*-Corrected	*rho*	*p*-Corrected	*rho*	*p*-Corrected
C3a	0.01	1.000	−0.21	0.768	−0.09	1.000	−0.10	1.000	0.10	1.000	−0.17	1.000	0.46	0.000
C5a	−0.02	1.000	−0.09	1.000	−0.06	1.000	−0.20	1.000	0.07	1.000	−0.11	1.000	0.19	0.828
C5b−9	−0.07	1.000	−0.33	0.081	−0.10	1.000	0.01	1.000	0.23	0.621	−0.17	1.000	0.41	0.000

## Data Availability

Data from the study reported here are available from the corresponding author on reasonable request.

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
