# Peer review of "Complement Activation Products in Patients with Chronic Schizophrenia"

_jcm, 2023, doi:10.3390/jcm12041577_

Round 1

Reviewer 1 Report

Title is inappropriate, as this rather represents a pilot study, does  not answering the question what role CC plays i  chronic schizophrenia

Abstract is not well structured and results are presented incorrect(in abstarct) and partly later in discussion ), as there is only 1 significant result in even a small sample. The exact C3a result including ranges and lab units should be noticed in the abstract.

Intro: could be more clear organized.

Methods: lab units are not written nor the firms from which the lab methods were used or if self-developed, then how, needs detailed description  nor are units named  in table

Results in principle ok, but language too broad.

Discussion: not well organized, consider an additional table with an overview on previous studies and their contradictory findings, and discuss on that basis the findings of this study  about by including consideration of the possible differences of lab methods  influencing the results

structure of discussuion should be improved

Beyond improving  language also the number of spelling errors should be eliminated.

Reviewer 2 Report

The authors present an interesting study where they undertook complement activation marker quantification in chronic schizophrenia patients. The scientific approach is sound and the research question is interesting. This is a timely study contributing to the overall knowledge of the biological mechanisms of schizophrenia and the involvement of the complement system.

I have some minor comments:

The novelty of your findings in context with previous studies is unclear and needs to be presented better in the abstract, introduction and discussion/conclusions.

Overall, the introduction seems disjointed. While information is presented, the introduction could be more logically structured.

Further minor points:

1)    Introduction: should also introduce the current evidence of complement involvement in psychosis development and schizophrenia. For instance, proteomics analyses identified the complement system in the development of psychosis, and Sekar et al (2016), identified MHC III and complement C4 variant as a risk factor associated with schizophrenia.

2)    Line 73, spelling: “release of or C5a and”

3)    Line 75, C4a is not strictly speaking an anaphylatoxin and a definite function has not been defined yet. (see doi: 10.1159/000371423 for reference)

4)    Line 78-79: specific references required for statements

5)    Line 86: please add: complement receptor 3 (CR3; CD11b/CD18) binds with C3 degradation fragment iC3b.

6)    Line 87-88: rephrase and clarify this sentence. Is there specific evidence for a joint mechanism?

7)    Line 90-91: cite further proteomic studies providing evidence for altered C3 or C5.

8)    Line 94-95. Relevance in this context? May move up to the beginning of the introduction, although the focus should be on neuroinflammatory or neurodegenerative conditions. Or clearly explain the link of SLE to neuropsychiatric manifestation.

9)    Line 104: relevance?

10) Line 107-114: relevance? Either make the relevance with regards to your research question clearer or remove.

11) Line 117: please clarify: complement components. I would consider replacing the wording complement components with complement activation products. A clearer introduction to the utility of these activation markers to support evidence of complement activation would be helpful. Also, a clearer introduction to why these specific markers were chosen.

12) Measuring 3 activation markers is not a comprehensive analysis of complement activation. One could argue that these are inflammatory markers as well as a marker of overall complement activation.

13) Define the aim of your study in the introduction

14) Line 132: define the abbreviation “SUD”

15) Line 151, please define whether plasma or serum was used, if plasma-please give information of EDTA-plasma etc. Also include information about storage conditions of the samples (-80C?) and length of storage (months/years?) before the assay. Were the samples split into aliquots and did they undergo repeated freeze-thaw cycles? This is important as complement activation fragment levels are affected by processing and storage conditions and freeze/thaw.

16) Line 151-154: Please include product codes for the ELISA kits.

17) Line 195. Please state accurate p value or e.g. p<0.00001 

18) Table 2, what units were measured for C3a etc?

19) Have you considered the immunological effects of antipsychotics and why was that not included in the analysis of covariance?

20) See also line 230 with regards to the impact of antipsychotic drugs on inflammatory markers. This may need further discussion.

21)  It would be useful to add a Figure with graphs (e.g., a box plot) showing the sample distribution of C3a, C5a and C5b-9 in SCH vs HC: Differential complement activation product levels “between two groups examined with Student’s t test (if the relevant as- 164 sumptions were met) and the Mann-Whitney U test (if the relevant assumptions were not 165 met) “

22) Line 251-252: relevance? remove.

23) Line 264: spelling

24) Line 296: spelling

25) Discussion: clarify the novelty of your findings in the context of other studies measuring the same activation fragments in similar cohorts. This also needs to be reflected in the abstract and when stating your research question and aims in the introduction. At the moment, the novelty of your findings in the context of other work is not clear enough.

26) Reference (8), while the link to autoimmunity is interesting, it does not add to this paper in the context of complement.

27) Reference (15), consider replacing with: https://doi.org/10.3389/fimmu.2015.00262

Round 2

Reviewer 1 Report

To interpret this pilot study findings a "predictor of schizophrenia" is inappropriate
